# Elevated Expression of HSP72 in the Prefrontal Cortex and Hippocampus of Rats Subjected to Chronic Mild Stress and Treated with Imipramine

**DOI:** 10.3390/ijms25010243

**Published:** 2023-12-23

**Authors:** Adam Bielawski, Agnieszka Zelek-Molik, Katarzyna Rafa-Zabłocka, Marta Kowalska, Piotr Gruca, Mariusz Papp, Irena Nalepa

**Affiliations:** 1Department of Brain Biochemistry, Maj Institute of Pharmacology, Polish Academy of Sciences, Smętna 12, 31-343 Kraków, Poland; bielaw@if-pan.krakow.pl (A.B.); zelek@if-pan.krakow.pl (A.Z.-M.); zablocka@if-pan.krakow.pl (K.R.-Z.); marcik48@op.pl (M.K.); 2Behavioral Pharmacology Laboratory, Maj Institute of Pharmacology, Polish Academy of Sciences, Smętna 12, 31-343 Kraków, Poland; gruca@if-pan.krakow.pl (P.G.); nfpapp@cyfronet.pl (M.P.)

**Keywords:** chronic mild stress, depression, imipramine, heat shock protein, HSP90, inducible HSP90A, constitutive HSP90B, HSP70, inducible HSP72, constitutive HSC70

## Abstract

The HSP70 and HSP90 family members belong to molecular chaperones that exhibit protective functions during the cellular response to stressful agents. We investigated whether the exposure of rats to chronic mild stress (CMS), a validated model of depression, affects the expression of HSP70 and HSP90 in the prefrontal cortex (PFC), hippocampus (HIP) and thalamus (Thal). Male Wistar rats were exposed to CMS for 3 or 8 weeks. The antidepressant imipramine (IMI, 10 mg/kg, i.p., daily) was introduced in the last five weeks of the long-term CMS procedure. Depressive-like behavior was verified by the sucrose consumption test. The expression of mRNA and protein was quantified by real-time PCR and Western blot, respectively. In the 8-week CMS model, stress alone elevated HSP72 and HSP90B mRNA expression in the HIP. HSP72 mRNA was increased in the PFC and HIP of rats not responding to IMI treatment vs. IMI responders. The CMS exposure increased HSP72 protein expression in the cytosolic fraction of the PFC and HIP, and this effect was diminished by IMI treatment. Our results suggest that elevated levels of HSP72 may serve as an important indicator of neuronal stress reactions accompanying depression pathology and could be a potential target for antidepressant strategy.

## 1. Introduction

At the cellular level, the stress reaction, including the heat shock protein (HSP) response, is a highly conserved cellular pathway activated after exposure to noxious stimuli to promote cell survival. HSP was first described by Ferrucio Ritossa in 1962 as a gene induced by heat shock stress to deal with denaturing folded proteins [1]. Currently, HSPs are recognized as molecular chaperones that assist in the correct folding of nascent and stress-accumulated proteins. HSPs, together with proteins that regulate their activity and proteins assisting in the process of removing misfolded and aggregated proteins, create the network of proteins responsible for maintaining cellular protein homeostasis [2]. In line with the new alphanumeric classification of HSP, there are five main families of HSP: HSPH (previously HSP110), HSPC (previously HSP90), HSPA (previously HSP70), DNAJ (previously HSP40) and HSPB (previously small HSP) [3], described in detail elsewhere [4]. HSPA and HSPC members are structurally and functionally similar HSPs, and impairments in one family were shown to affect the expression and activity of the second family (e.g., [5]). HSPA and HSPC members bind substrates via the middle binding domain, and their chaperone activity depends on ATP hydrolysis in the N-terminal domain [2]. Both families contain two types of members: constitutive (cognate) members, which perform housekeeping functions, and inducible members, which are induced rapidly by stress. Most HSPA and HSPC members show predominant cytosolic localization with the possibility of shuttling between cellular compartments [4,6] and therefore play a central role in protein translation, post-translational modifications and thus effective intracellular signaling. HSPA and HSPC coordinate chaperone activities across cellular compartments and cooperate in the folding process [7].

Deficiencies in chaperone machinery may lead to the development or progression of protein-misfolding diseases, e.g., neurodegenerative diseases, type 2 diabetes, peripheral amyloidosis, lysosomal storage disease, cystic fibrosis, cancer and cardiovascular disease (see [8]). Clinical evidence has revealed that HSP alterations also accompany depression [9].

Laboratory research indicates that pharmacological modulation of intracellular levels of HSPs, particularly HSPA, may have therapeutic benefits in treating diseases where chaperone function is impaired [10,11,12,13,14]. The molecular mechanism underlying chaperone system impairments in stress-related psychiatric disorders is far from understood in terms of both chaperone pathology identification and therapeutic options. Our previous study in a mouse model of depression revealed that early stress exposure evokes long-term downregulation of mRNA for stress-inducible members of HSPA and HSPC in the PFC, accompanied by a decreased density of alpha1B adrenoceptors [15].

The current study aimed to assess the mRNA and protein expression of stress-induced HSPA and HSPC members, HSP1A (also known as HSP72) and HSPC1 (HSP90A), as well as constitutive HSPA8 (HSC70) and HSPC3 (HSP90B), in the prefrontal cortex (PFC), hippocampus (HIP) and thalamus (Thal) of rats subjected to the chronic mild stress (CMS) procedure performed for 3 and 8 weeks (Figure 1). Moreover, in prolonged CMS, we checked whether pharmacotherapy with the antidepressant drug imipramine (IMI) could normalize CMS-evoked HSP changes. To confirm whether changes in mRNA levels are related to the cytosol compartment, we measured the cytosol level of proteins for selected HSPA and HSPC members.

The chosen brain structures PFC, HIP and Thal have been shown to send major glutamatergic afferents to the nucleus accumbens that are highly affected by neuronal atrophy in stressed animals displaying anhedonia [16,17]. Moreover, these structures are known to control emotion, learning and memory, the integration and transmission of stimuli and attention, which seem to be involved in stress-sensitive mechanisms underlying major depression.

## 2. Results

### 2.1. Evaluation of Sucrose Consumption in Stress-Reactive vs. Stress-Nonreactive Animals and in Imipramine-Responding vs. Imipramine-Nonresponding Animals: Generating Experimental Groups for Biochemical Studies

In line with previous studies, the exposure to several weeks of CMS led to the development of an anhedonic-like phenotype, manifested by a reduction in sucrose consumption. Two cohorts of animals were used in the current experiments. The first was for 3 weeks of CMS with three experimental groups (Figure 1A), and the second was for 8 weeks of CMS with five animal groups (Figure 1B). In the latter cohort, a treatment with imipramine for the last 5 weeks has been included for some groups, as depicted in Figure 1B (left panel).

In animals exposed for 3 weeks to CMS and showing vulnerability to stress, the last sucrose intake score was significantly reduced (F(2, 15) = 15.51, *p* < 0.001), and the score values were 4.4 ± 1.1 vs. 19.7 ± 3.2 (*p* < 0.001), for the stress-reactive group and the controls, respectively. In the stress-nonreactive animals, sucrose intake was comparable to that in the sham group (15.6 ± 0.09 vs. 19.7 ± 3.2) (Figure 1A, middle panel).

In animals exposed for 8 weeks to CMS, the experimental groups were selected on the basis of the last sucrose intake score (F(4, 25) = 14.54, *p* < 0.001). The sucrose intake was reduced for the stress-reactive/sal compared to sham/sal group (7.7 ± 1.1 vs. 14.6 ± 1.6, *p* < 0.001, respectively). In the group with stress-induced anhedonia and responding to IMI treatment, the sucrose intake score returned to the control level (13.9 ± 0.7 vs. 14.6 ± 1.6), while in animals with stress-induced anhedonia and not responding to IMI treatment, the sucrose intake score did not return to the control level, 5.1 ± 1.1 vs. 14.6 ± 1.6, *p* < 0.001) (Figure 1B, middle panel).

### 2.2. Effects of Three Weeks of CMS on the mRNA Expression of HSP70 and HSP90 Family Members

One-way ANOVA followed by Tukey’s test showed a slight decrease in HSP90A mRNA expression of inducible HSP90A in the HIP of stress-reactive rats vs. sham-group rats (by 12%; F(2,15) = 5.28, *p* < 0.05). No other changes in the mRNA expression of genes belonging to the HSP70 and HSP90 families in the PFC, HIP and Thal were observed (Table 1).

### 2.3. Effects of Eight Weeks of CMS on the mRNA Expression of HSP70 Family Members

One-way ANOVA for the mRNA expression of inducible HSP72 did not show significant effects in the PFC (F(4,25) = 1.42, *p* > 0.05), HIP (F(4,25) = 1.79, *p* > 0.05) or Thal (F(4,25) = 1.05, *p* > 0.05). Therefore, to statistically compare differences among treatment groups, we performed contrast analysis. In the PFC, contrast analysis revealed more abundant expression of inducible HSP72 mRNA in the stress/IMI nonresp group than in the stress/IMI resp group (by 82%, *p* < 0.05) (Figure 2A). In the HIP, we noted increased expression of HSP72 mRNA in the stress/IMI nonresp group vs. the stress/IMI resp group (by 59%, *p* < 0.05) and vs. sham/sal control group (by 57%, *p* < 0.05) (Figure 3A).

Neither one-way ANOVA nor contrast analysis showed any changes among groups in the mRNA expression of constitutive HSC70 in the PFC, HIP (Appendix A) and Thal (Appendix A).

### 2.4. Effects of Eight Weeks of CMS on the mRNA Expression of HSP90 Family Members

In the HIP, one-way ANOVA for the mRNA expression of cognate HSP90B showed a main effect of treatment (F(4,25) = 3.19, *p* < 0.05). Furthermore, the planned contrast comparison showed increased expression of HSP90B mRNA in the CMS group compared to the sham/sal group (by 16%, *p* < 0.05). This CMS-evoked elevation of mRNA was reversed by IMI treatment (*p* < 0.05 between groups stress/sal and stress/IMI resp) (Figure 4A). A lack of changes in the expression of inducible HSP90A mRNA in the HIP among groups was observed (Appendix A). We did not observe any changes in the mRNA expression of inducible Hsp90A and constitutive Hsp90B in the PFC (Appendix A) and Thal (Appendix A), respectively.

### 2.5. Effects of Eight Weeks of CMS on the Cytosolic Protein Expression of HSPs

One-way ANOVA did not show a main effect of treatment in either the PFC (F(4, 25) = 1.27, *p* = 0.05) or the HIP (F(4, 25) = 1.53, *p* > 0.05). However, planned comparison followed by contrast analysis showed that CMS evoked an increase in HSP72 protein expression in the cytosol compartment in the PFC and HIP by 49% and 41%, respectively, vs. the sham/sal group (Figure 2B and Figure 3B, Appendix A). Contrast analysis of the cytosolic level of constitutive HSP90B protein revealed no changes among groups in the HIP (Figure 4B, Appendix A).

### 2.6. HSP72 and HSP90B Colocalize with Neuronal Cells

Immunofluorescence analysis showed double staining for HSP72 and the neuronal marker NeuN in the PFC and HIP areas (Figure 5A–C,G–I). Likewise, double immunofluorescence staining for Hsp90B and the neuronal marker NeuN was observed in the PFC and HIP areas (Figure 5D–F,J–L). In addition, corresponding images with the DNA-binding DAPI dye are included in the Appendix A and show cell nuclei differentiated by labeling with the additional marker other than NeuN.

Furthermore, no double immunofluorescence staining was found for HSP72 and HSP90B or for the astrocyte marker GFAP in either the PFC or HIP areas (Appendix A).

## 3. Discussion

The main finding of the study is that depressive-like behavior evoked by long-term exposure to CMS is accompanied by the induction of HSP72 in the rat PFC and HIP. Moreover, that increase is related to neuronal cells and seems to be involved in the process of resistance to antidepressant therapy.

### 3.1. Applicability of Experimental Groups Generated in the CMS Model

CMS exposure leads to the development of depression-like symptoms in animals, which we confirmed in our study. In this model, rats subjected for a prolonged period of time to a variety of mild stressors gradually decrease their responsiveness to rewarding stimuli, which is comparable to anhedonia, the core clinical symptom of the melancholic subtype of major depressive disorder (reviewed by [18]). After 3 weeks of CMS exposure, in addition to the generated stress-reactive group, we obtained a “stress nonreactive” group that failed to reduce their drinking of a palatable solution despite being under the influence of stress. Here, we utilized this group to evaluate whether changes in the expression of HSP72, HSC70, HSP90A and HSP90B underlie anhedonic resilience. In the experiment with 8 weeks of CMS, for the last 5 weeks, we administered IMI, a classic antidepressant drug with a tricyclic chemical structure, to rats as a reference drug widely used in pharmacological studies looking for the mechanisms underlying major depression. We observed that IMI reversed anhedonia in the “stress IMI resp” group of rats; however, we also differentiated a “stress IMI nonresp” group in which IMI applied for 5 weeks did not reverse anhedonia. The group that failed to respond to IMI treatment, the “stress IMI nonresp” group, is in agreement with observations of other laboratories [18,19], and it is believed to reflect patients with treatment-resistant depression [20].

### 3.2. Evaluation of the Effects of CMS and IMI Therapy on the Cerebral Expression of HSP

Changes in mRNA expression are widely regarded as indices of alterations in physiological functions (e.g., [21,22]). Additionally, in the case of HSPs, their gene structure and the regulation of transcription have been evolutionarily adapted to the rapid stress response with no delay between mRNA transcription and protein translation, e.g., the activity of HSF-1, the main transcription factor of HSPs, depends on direct binding to HSPA and HSPC [23]; the gene structure of HSP72 is without introns [24]. Based on the above, we assumed that the mRNA expression of HSPA and HSPC members corresponded to the total cellular protein level synthesized in the studied brain structures. In turn, the Western blot data described here reflect the level of proteins present only in the cytosol fraction. Although HSC70, HSP72, HSP90A and HSP90B are predominantly localized in the cytosol, these proteins are also present in other cellular fractions [4,25] that can be affected by CMS. The comparison of CMS-evoked changes in mRNA and cytosolic protein levels allowed us to better understand the cellular loci of the stress response involving HSPA and HSPC proteins. To the best of our knowledge, this is the first report showing the effects of CMS exposure and IMI treatment on the expression of HSP72, HSC70, HSP90A and HSP90B in stress-sensitive brain structures (PFC, HIP and Thal).

#### 3.2.1. HSPA Family

The stress-evoked HSP72 and constitutive HSC70 members of the HSPA family studied here belong to the first cytosol molecular chaperones, which aid in the folding or refolding of newly synthesized cellular proteins and transporting them to appropriate cellular compartments [26]. During a cellular stress response, cytosolic HSPA cooperating with HSPC was shown to prevent cell death by refolding misfolded proteins after exposure to high temperature, cytokines, alcohols, heavy metals, radiation, metabolites or free radicals [27]. In our study, we noted an augmented cytosolic level of HSP72 in the PFC and HIP of rats exposed to long-term (8-week) CMS, which was normalized by antidepressant therapy with IMI. It is worth adding that in the HIP, the increase in cytosolic HSP72 levels in the CMS group was accompanied by augmented mRNA synthesis. In the PFC, however, CMS did not affect the mRNA level of HSP72. We suspect that the HSP72 increase in the PFC was due to the rearrangement of the amount of this protein between fractions, which was shown to occur, particularly under cellular stress [4]. In general, the induction of HSP72 in the cytosol enhances the ability of stress cells to cope with an increased amount of unfolded/denatured proteins [28]. Therefore, we think that the increased cytosolic level of HSP72 seen under CMS reflects the cellular refolding stress response. Accordingly, in experimental models, HSP72 overexpression or upregulation was shown to protect against stress-induced damage to the cell [10], reduce neurological injuries [11,12,29,30] and alleviate the process of neurodegeneration [13]. The induction of HSP72 in HIP slices reduced the vulnerability of CA1 to glutamate-dependent neurotoxicity [31], while in the PFC, it prevented the long-term serotoninergic neurotoxicity evoked by 3,4-methylenedioxymethamphetamine (MDMA, ecstasy) administration [32]. On the other hand, increased activity of HSPA is known to accompany disease states. A clinical study identified an increased level of HSP72 as a risk factor for depression [33]. Additionally, increased interaction of HSP72 with glucocorticoid receptor (GR) in the lymphocytes of bipolar patients was described [9]. Our results, performed using a validated preclinical model of depression, agree with clinical observations and indicate that a pathological increase in the cytosolic level of HSP72 within the PFC and HIP, likely connected to an impaired protein refolding process, can be resolved by IMI. It should be noted that IMI alone, in our study, did not influence the HSP level. However, alleviated by CMS, the cytosolic level of HSP72 in the PFC and HIP indicated that this chaperone is an important target for classic antidepressant pharmacotherapy. The literature data showing the effects of chronic IMI treatment on HSP expression are different from ours. Elakovic et al. showed that HSP72 and HSP90A levels were decreased in the cortex and increased in the HIP of male rats [34]. The reason for this discrepancy is likely a different regimen of IMI administration. Although the route of treatment, dose per day and time of decapitation after the last dose were the same in both cases, in our model, IMI was administered for 5 weeks, which was much longer than in the study of Elakovic (3 weeks). The IMI treatment we applied was chosen based on previous data showing a maximal anti-anhedonic effect in a rat model of CMS after 5 weeks of IMI application and no effect after 3 weeks of IMI [35]. The lack of changes in HSP levels after the administration of IMI alone that we have shown in our study allows us to ensure that the results we present are specifically related to the mechanisms of depression and the effectiveness of antidepressant treatment generated by CMS molecular changes.

Although IMI treatment in our study normalized the cytosolic level of HSP72, its mRNA level was augmented in the PFC and HIP of rats resistant to antidepressant therapy. These results suggest that increased synthesis of HSP72 plays an essential role in patients’ resistance to antidepressant treatment. Regarding the potential engagement of HSPA in the cellular mechanism underlying resistance to antidepressants, it should be mentioned that in addition to protein folding/refolding functions in the cytosol, HSPA is engaged in the activation of protein degradation pathways, specifically oxidative stress reactions and the unfolded protein response [4]. These processes initiated in mitochondria and the endoplasmic reticulum (ER) were shown in preclinical and clinical studies. For example, in animal models of unadaptable stress, elevated HSP 72 levels were accompanied by increased amounts of free radicals and oxidative stress reactions in the brain [36,37]. Moreover, clinical studies have shown a significant relationship between depressive episodes and oxidative stress markers in blood serum [38]. A clinical study performed in depressed patients also evidenced the presence of ER stress and the mediation of the unfolded protein response in the brains of suicidal patients [39].

Another possible explanation for the increase in the expression of HSP72 presented here may be the activation of apoptotic processes in stress-impaired cells. It seems consistent that CMS increases the number of caspase-3-positive neurons in the cerebral cortex [40]. Others have shown that HSP72 might protect hippocampal neurons from the apoptosis induced by chronic psychological stress in mice [41]. High expression of inducible HSP72 protects cells against apoptosis and favors cell survival [10].

The study showed that the elevation of HSP72 was abolished by chronic treatment with IMI, which may silence cellular stress reactions connected with apoptosis processes. Increased proapoptotic serum activity in patients with major depression was shown [42]. Additionally, accelerated apoptosis of blood leukocytes was found in the blood of patients with major depression [43,44]. Others have indicated that the antidepressant desipramine reverses this proapoptotic effect of CMS [40]. Neuroprotection by imipramine against apoptosis in hippocampus-derived neuronal stem cells through the activation of the neurotrophic factor BDNF and the MAPK kinase pathway has been shown recently by others [45,46]. Additionally, antidepressant drugs, fluoxetine, reboxetine, tranylcypromine and electroconvulsive seizures upregulated the expression of BCL-2 family members in selected brain structures of rats [47]. Desipramine could also activate BCL-2 expression and inhibit apoptosis in hippocampus-derived adult neuronal stem cells [48]. Another antidepressant drug, moclobemide, also upregulated Bcl-2 expression and induced neuronal stem cell differentiation [49].

Adrenergic receptors involved in the action of antidepressant drugs are mediators of stress induction of extracellular HSP72 in the plasma of rats [50]. Additionally, the intracellular level of HSP72 is upregulated via the alpha-1 adrenoceptor pathway [51,52,53,54]. Moreover, our previous paper showed that brief maternal separation affected brain α1-adrenoceptors and apoptotic signaling in adult mice [15]. In addition, others have shown the involvement of the adrenergic pathway in HSP72 induction in rats [55]. This may be the possible mechanism of the neuroprotective effects of some antidepressant drugs with an adrenergic component of action.

The lack of changes in the expression of constitutive HSPA shown in our study might have been associated with the fact that this protein assists mainly in the process of protein synthesis during normal physiological conditions and takes part in the response to stress agents. In addition to the role of constitutive members of HSPA in nascent protein folding, the increased level of HSC70 was also shown to accompany autophagy in an experimental model of neurodegeneration [56]. The lack of CMS-evoked alterations in the expression of HSC70 in our study may suggest that the process of autophagy does not underlie the mechanisms of depression.

#### 3.2.2. HSPC Family

Evoked by stress, HSP90A and constitutive HSP90B are cytosolic HSPC chaperones. They both assist HSPA chaperones in the processes of protein folding that take place in the cytosol [4]. In response to stress and elevated corticosterone levels, HSPC chaperones in cooperation with HSPA were shown to associate with GR, translocate GR into the nucleus and thus regulate the transcriptional activity of this receptor [57,58,59,60]. During stress reactions, HSPC may itself bind to a multichaperone complex regulating Heat Shock Factor-1 (HSF1), which is the transcription factor for HSPA and HSPC genes [61]. In our study, three weeks of exposure to CMS slightly decreased HSP90A mRNA expression in the HIP of rats. Similar HSP90A downregulation was noted previously in a mouse stress model of neonatal separation from the mother [15], where stress-evoked deregulation of GR signaling was suggested. Since HIP has been recognized as the gateway to remodel brain structure and function by exposure to stress hormones and the main role of HSP90A chaperone is to regulate protein folding processes, our result may reflect stress-evoked disturbances in protein folding processes known to be impaired by neurodegeneration. Interestingly, we showed that long-term (8 weeks) CMS enhanced the expression of constitutive HSP90B in the HIP and that IMI treatment abolished this increase. Alterations in the level of the constitutive form of HSPC after stress exposure seem to be typical HSPC responses to prolonged stress because others have already shown that the expression of constitutive HSP90B in the brain is lower and more regulated in the brain than that of inducible HSP90A, in contrast to peripheral organs [62]. The observed increase in HSP90B mRNA expression is consistent with the result obtained for HSP72 and might be a response to the stress reaction caused by apoptotic processes because HSP90 overexpression was also shown to inhibit apoptosis [63]. Additionally, HSP90 has the ability to inhibit apoptosis as a result of a negative effect on the proapoptotic function of APAF-1 [63]. The increase in HSP90B expression may also be connected with the regulation of the glucocorticoid response to chronic stress [58]. Eventually, the normalization of the HSP90 mRNA level in the HIP of rats responding to imipramine treatment might be involved in the control of neurotransmitter release from synaptic ends [64].

#### 3.2.3. HSP as a Potential Target for Antidepressive Treatment

HSPA and HSPC are part of the GR chaperone complex responsible for the binding of GR in the cytosol, the nuclear translocation and mobility of GR, DNA binding and clearance (see e.g., [25,65]). Therefore, the modulation of HSP activity may regulate downstream corticosteroid signaling, which was shown to be impaired in stress-sensitive brain structures of depressive patients and observed in preclinical studies including ours (e.g., [66,67,68,69]). It has been proven that experimental hyperacetylation of HSPC, which modulates intracellular GR chaperone complex mobility, triggers stress resilience in mice models of PTSD ([65]). Depression impairs mechanisms of the inflammatory response [70], highly dependent on GR signaling (and therefore GR chaperone complex activity) and/or HSP72 direct activation [71,72]. We demonstrated previously that the anhedonic-like phenotype evoked by CMS is associated with neuroinflammation, sustained by the increased expression of proinflammatory cytokines IL-1 and IL-6 accompanied by microglial activation [73]. In our previous studies, we showed lower IL-1β mRNA expression in PFC and HIP of CMS rats not responding to IMI vs. CMS rats responding to IMI [74]. The upregulation of Hsp72 (current study) we found in PFC and HIP of only the “stress/IMI nonresp” group could be associated with the CMS-induced impairment of inflammatory response, mentioned above, and observed also in patients not responding to antidepressants [70]. Depression is known also to affect the neuronal cell death and viability processes which we observed previously in the PFC and HIP of animals in anhedonia-related stress models, e.g., [15,75]. HSPA and HSPC are upstream regulators of apoptosis signaling [76,77,78], and our data suggest they are engaged in the anhedonia-evoked apoptosis in PFC and HIP [15].

In Figure 6, we indicate possible downstream targets of HSP72, whose mRNA expression is increased in the PFC and HIP of the CMS rats that did not respond to imipramine, as revealed by the current study.

### 3.3. Localization and Dynamics of CMS-Induced Changes in HSP Expression

Our immunohistochemical data revealed neuronal expression of HSP72 and HSP90B in the PFC and HIP, which agrees with other observations [31] and suggests that the HSP changes described here could be directly related to the altered function of the brain circuitry observed in human psychiatric disorders [82,83,84]. The HIP–PFC connection is recognized as a critical pathway for stress pathology [85], and neuronal plasticity was shown to be affected in these structures by CMS [86,87,88] and by unadaptable stress models [83,89]. Our study revealed a similar increase in HSP72 in the PFC and HIP of rats exposed to 8 weeks of CMS, which was accompanied by depressive-like behavior. We cannot be sure from our data the exact role of HSP72 upregulation in neuronal structure impairment, but we know that HSP72 is engaged in protein refolding processes and cell death regulation. Therefore, it is tempting to speculate the involvement of HSP72 upregulation in the mechanism of dendrite atrophy and the reduced volume of hippocampal and frontal cortical brain regions noted in major depression patients [90,91,92].

Comparing the effects of different durations of exposure to CMS, we noted the presence of changes in HSP levels after long-term CMS, lasting for 8 weeks, but not after 3 weeks of CMS. Studies in animal stress models of depression have revealed bidirectional effects on neuronal remodeling (related mainly to the glutamatergic system) that depend on stressor strengths and the time of exposure to stress. Based on the above, acute or short-term stress triggers homeostatic, adaptive changes in brain structures and function. However, chronic stress models are considered models of psychiatric disorders in which unadaptable stress reactions are demonstrated [93,94]. Therefore, the changes in HSP levels we observed after 8 weeks of CMS accompanied by depressive-like behavior and reversed by IMI treatment, at least in the group of treatment responders, underlie the mechanism of depression.

## 4. Materials and Methods

### 4.1. Animals

Experiments were conducted on male Wistar Han rats purchased from Charles River, Germany. The animals weighed approximately 300 g upon arrival and approximately 300 g at the start of the stress procedure. Before the start of behavioral experiments, rats were adapted to laboratory conditions for 1 month. The animals were housed singly in plastic cages (40 × 25 × 15 cm) with food and water freely available in a standard 12 h/12 h light/dark cycle at a temperature of 22 °C. Exceptionally, grouping, food and/or water deprivation and changing the light/dark cycle were applied as stress parameters. The study was approved by the Local Ethical Commission for Animal Experiments at Maj Institute of Pharmacology, at the Polish Academy of Sciences, in Krakow (Permit No. 748/2010, date 22 April 2010).

### 4.2. Sucrose Consumption Test

All animals were first trained to consume the sucrose solution (1%) for 6 weeks, according to the procedure described earlier [95]. Every week of training, the sucrose solution was presented to rats for 1 h in their home cage after 14 h of food and water deprivation. The bottles’ weights were measured before and after each drinking session as the sucrose intake score. Subsequently, sucrose intake was monitored under similar conditions throughout the whole experiment once a week as an indicator of the stress effect (test of anhedonia), as previously demonstrated [19].

### 4.3. Chronic Mild Stress Protocol

The CMS procedure was performed as described previously ([19], ref as above). Briefly, the weekly stress regime was composed of two periods of food or water deprivation, a 45° cage tilt, intermittent illumination (lights on and off every 2 h), a soiled cage (250 mL water in sawdust bedding), paired housing, low-intensity stroboscopic illumination (150 flashes/min) and two periods of no stress stimuli. All stressors lasted for 10–14 h and were applied individually and continuously, day and night. Control animals were housed in a separate room without stressors.

#### 4.3.1. Stress-Reactive and -Nonreactive Animals

On the basis of their sucrose intake in the baseline test, the animals with stable sucrose solution consumption were randomly divided into two matched groups. One group of animals was subjected to CMS for a period of 3 weeks, while the nonstressed animals were housed in a separate room. After 3 weeks of stress, the animals were divided into two subpopulations: one subpopulation responded to CMS administration in the last behavioral sucrose test (named stress-reactive), and the second subpopulation did not respond to CMS (named stress-nonreactive) (see Figure 1A).

#### 4.3.2. Imipramine-Responding and -Nonresponding Animals

Similarly, on the basis of their sucrose intake in the baseline test, the animals with stable sucrose solution consumption were randomly divided into two matched groups. One group of animals was subjected to CMS for a period of 8 weeks, while the nonstressed animals were housed in a separate room. After the initial 3 weeks of stress, both stress-reactive animals and control animals were further divided into two subgroups, which, for the next 5 weeks, received an injection of saline (sal, 10 mg/kg, i.p., daily) or IMI (10 mg/kg, i.p., daily). Finally, the following experimental groups were selected on the basis of the last sucrose intake score: control animals, stress/sal—animals with stress-induced anhedonia (and with the sucrose intake score reduced), sham/IMI—animals treated with IMI and not stressed, stress/IMI resp—animals with stress-induced anhedonia and responding to IMI treatment (the sucrose intake score returned to the control level), and stress/IMI nonresp—animals with stress-induced anhedonia and not responding to IMI treatment (the sucrose intake score did not return to the control level) (see Figure 1B).

### 4.4. Drug Administration

IMI (Sigma-Aldrich, Darmstadt, Germany) was dissolved in physiological saline and given at a dose of 10 mg/kg. The drug and vehicle were administered intraperitoneally at a volume of 1 mL/kg (i.p.) in the morning.

### 4.5. Tissue Preparation

Rats were decapitated 24 h after the last sucrose test. Whole brains or brain structures were isolated immediately after killing. Then, brain structures for mRNA and protein assays were frozen in dry ice (at −78.5 °C).

### 4.6. Real-Time Analysis of HSP mRNA Levels

Real-time analysis of mRNA expression was performed as described previously [96]. Briefly, total RNA was extracted from the brain structures using an RNeasy Mini kit, and possible genomic DNA contamination was eliminated by means of an RNAse-Free DNase Set kit according to the manufacturer’s instructions (Qiagen, Hilden, Germany). The concentration was assessed spectrophotometrically (Implen NanoPhotometer, Munich, Germany) and reverse transcribed (1000 ng/sample of total RNA) using oligo d(T) primers with a High Capacity cDNA Reverse Transcription Kit (Applied Biosystems, Waltham, MA, USA) according to the manufacturer’s protocol.

Products of reverse transcription reactions were amplified by PCR using Gene Expression PCR Master Mix (Applied Biosystems, Waltham, MA, USA) with appropriate commercially available TaqMan Gene Expression Assays (HSP72: Rn00583013_s1, HSC70: Rn00821191_g1, HSP90A: Rn00822023_g1, HSP90B: Rn01511686_g1, Hprt1: Rn01527840_m1; Applied Biosystems, Waltham, MA, USA). A QuantStudio 12K Flex (Applied Biosystems, Waltham, MA, USA) detection system for quantitative real-time detection of PCR products was used. The cycle conditions were set as recommended by the manufacturer: after an initial 2 min hold at 50 °C and 15 min at 95 °C, the samples were cycled 40 times at 95 °C for 15 s and 60 °C for 1 min. The threshold value (Ct) for each sample was set in the exponential phase of PCR, and the standard curve method was used to analyze the data. Hypoxanthine-guanine phosphoribosyltransferase (Hprt) was used as a reference gene whose expression was observed at a constant level in all experimental groups of animals.

### 4.7. Immunoblot Analysis of HSP72 and HSP90B Protein Levels

Cytosolic protein analysis, performed by Western blot, was restricted to HSP72 (PFC, HIP) and HSP90B (HIP) measured in all groups from the 8-week CMS model. The selection of samples for the study of protein expression served to confirm whether changes in mRNA levels are related to the cytosol compartment. Therefore, the selection was a consequence of the obtained data on the mRNA expression of the studied HSPs.

Tissue samples were homogenized in a glass–Teflon homogenizer (Glass Col, Terra Haute, IN, USA) in a 500 µL volume of ice-cold homogenization buffer containing 10 mM TRIS (Bioshop, Burlington, ON, Canada; pH 8.0), 0.1 M EDTA (Amresco, Solon, OH, USA) and 0.1 M NaCl (Amresco, Solon, OH, USA); a complete set of protease inhibitors (Sigma-Aldrich, St. Louis, MO, USA); and a phosphatase inhibitor cocktail (Sigma-Aldrich). The homogenate was centrifuged at 8000× *g* for 5 min at 4 °C. After centrifugation, the supernatant was collected as the cytosolic fraction. Protein concentrations were determined by using a BCA protein assay kit (Bio-Rad Laboratories, Hercules, CA, USA).

The quantification of HSP72 and HSP90B proteins by Western blot was based on a protocol described previously. Briefly, 10 or 20 µg of protein (HSP90B and HSP72, respectively) was separated electrophoretically on an SDS-12% polyacrylamide gel (Amresco, Solon, OH, USA). Proteins were transferred to nitrocellulose membranes (Bio-Rad Laboratories). After incubation with a blocking buffer—2.5% nonfat dry milk with a 2.5% V fraction of bovine serum albumin (Amresco, Solon, OH, USA)—the blots were probed overnight at 4 °C with primary anti-HSP72 or anti-HSP90B antibody (1:1000 or 1:2000, Abcam, Cambridge, UK, cat. no. ab47455 and ab53497, respectively). Following incubation with the secondary antibody (peroxidase-conjugated anti-mouse IgG, 1:4000, Roche Diagnostic, Rotkreuz, Switzerland), immunocomplexes were visualized by chemiluminescence using the Lumi-LightPLUS Western Blotting Kit (Roche Diagnostic, Rotkreuz, Switzerland) according to the manufacturer’s instructions. The quantification of the immunoblots was performed using FujiLas-1000 and MultiGauge V3.0 software (FujiFilm, Tokyo, Japan). The loading of protein for each lane was verified using α-tubulin or β-actin (1:5000, Santa Cruz Biotechnology, Inc., Santa Cruz, CA, USA; cat. no. sc-5286 and sc-47778, respectively) antibodies. All samples were normalized to their respective internal control protein.

### 4.8. Immunofluorescence Analysis

After isolation, rat brains were fixed in 4% paraformaldehyde (PFA) overnight. Then, the tissues were embedded in paraffin and coronally sectioned (7 µm) on a rotary microtome (Leica, Wetzlar, Germany) for different brain areas (hippocampus, prefrontal cortex; bregma between −1.30 mm and −4.30 mm) as described previously [59]. Chosen sections from corresponding regions of brains in animals were incubated overnight at 4 °C with primary anti-HSP72 (1:500, Abcam, Cambridge, UK, cat. no. ab47455) or anti-HSP90B (1:200, Abcam, Cambridge, UK, cat. no. ab53497) together with anti-NeuN (1:500, Abcam, Cambridge, UK, cat. no. ab177487) antibodies. Antigen-bound primary antibodies were visualized with anti-mouse Alexa-488 and anti-rabbit Alexa-594 coupled secondary antibodies (Molecular Probes, Eugene, OR, USA). Stained sections were analyzed and acquired by means of a fluorescence microscope (Nikon, Eclipse50i, Tokyo Japan) fitted with a camera and appropriate software (NIS Element, ver. BR 3.0).

### 4.9. Statistical Analysis

Statistical analysis of the results was performed with STATISTICA 12.0 software (StatSoft, Tulsa, OK, USA) using one-way analysis of variance followed by Tukey’s test. The normality of the distribution of variables and the homogeneity of variances were checked by Levene’s test. Additionally, in the CMS 8-week model, five treatment groups were generated, and planned comparisons with contrast analysis were performed. Three types of contrast were tested: stress/sal or sham/IMI vs. sham/sal, stress/IMI resp or stress/IMI nonresp vs. stress/sal and stress/IMI nonresp vs. stress/IMI resp or sham/sal groups. *p* values lower than 0.05 were regarded as statistically significant.

## 5. Conclusions

The data showed that long-term CMS is related to increased cytosolic HSP72 in the PFC and HIP, which can be normalized by IMI treatment. That effect, likely connected to the augmented refolding mechanisms occurring, is reversed by IMI in a way that is not dependent on its therapeutic efficacy. However, CMS evoked an increase in HSP72 mRNA synthesis in the HIP, which was not limited to the cytosol level of this protein and was still present in the group of rats resistant to IMI treatment. Our data indicate that CMS evoked an increase in the neuronal expression of HSP72, likely connected to apoptotic processes, which may be a crucial intracellular goal for treating drug-resistant depression.

## Figures and Tables

**Figure 1 ijms-25-00243-f001:**
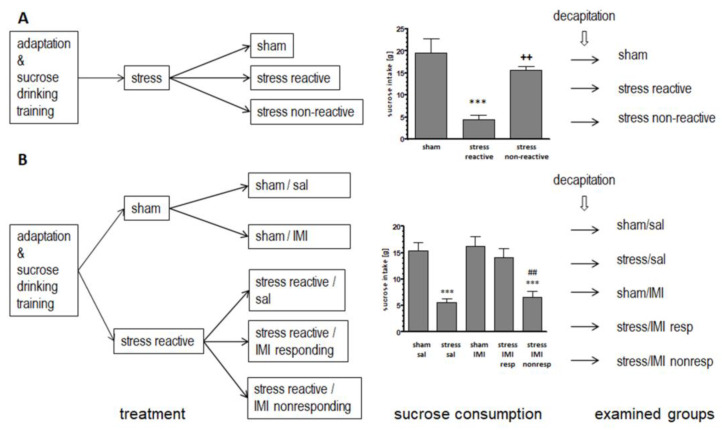
Schematic showing the experimental groups and consumption of 1% sucrose solution in rats exposed to CMS (stress) for (**A**) 3 weeks and (**B**) 8 weeks and simultaneously treated with IMI for the last 5 weeks (IMI, 10 mg/kg, ip). Vertical lines represent the standard error of the mean, n = 6; *** *p* < 0.001 relative to the sham or sham/sal group; ++ *p* < 0.01 relative to the stress-reactive group; ## *p* < 0.01 relative to the stress/IMI resp group; reactive—group of animals responding to CMS in the behavioral test.

**Figure 2 ijms-25-00243-f002:**
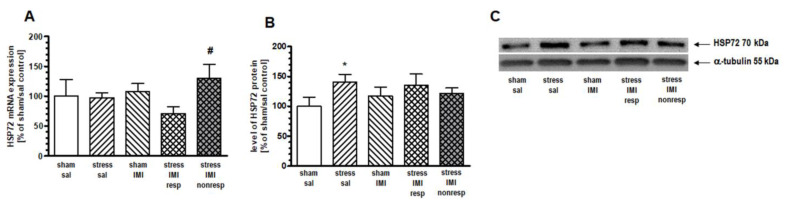
Levels of (**A**) mRNA and (**B**) protein expression of HSP72 in PFC of rats subjected to the CMS (stress) for 8 weeks and simultaneously treated with IMI (10 mg/kg, ip) for the last 5 weeks. Results were determined by real-time PCR and Western blot methods. Rats were sacrificed 24 h after the last administration of IMI or saline (sal). Results are expressed as a percentage of sham/sal group ± standard error of mean (SEM); n = 6; (**C**) representative blots; * *p* < 0.05 relative to sham/sal, # *p* < 0.05 relative to stress/IMI resp group; stress/IMI resp—group of CMS rats that responded to IMI treatment (reversed anhedonia in sucrose test); stress/IMI nonresp—group of CMS rats that did not respond to IMI.

**Figure 3 ijms-25-00243-f003:**
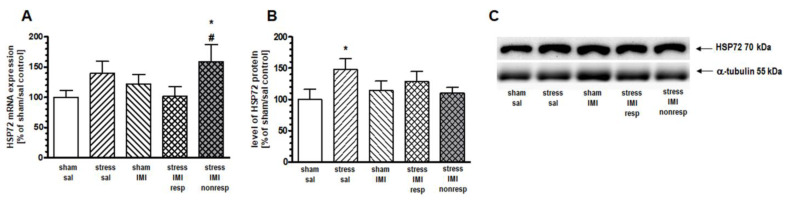
Levels of (**A**) mRNA and (**B**) protein expression of HSP72 in HIP of rats subjected to the CMS (stress) for 8 weeks and simultaneously treated with IMI (10 mg/kg, ip) for the last 5 weeks. Results were determined by real-time PCR and Western blot. Rats were sacrificed 24 h after the last administration of IMI or saline (sal). Results are expressed as a percentage of sham/sal group ± standard error of mean (SEM); n = 6; (**C**) representative blots; * *p* < 0.05 relative to sham/sal, # *p* < 0.05 relative to stress/IMI resp group; stress/IMI resp—group of CMS rats that responded to IMI treatment (reversed anhedonia in sucrose test); stress/IMI nonresp—group of CMS rats that did not respond to IMI.

**Figure 4 ijms-25-00243-f004:**
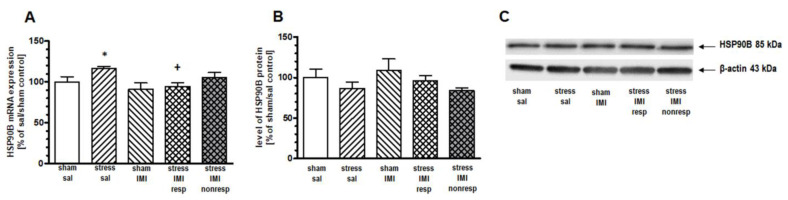
Levels of (**A**) mRNA and (**B**) protein of HSP90B in HIP of rats subjected to the CMS (stress) for 8 weeks and simultaneously treated with IMI (10 mg/kg, ip) for the last 5 weeks. Results determined by real-time PCR and Western blot. Rats were sacrificed 24 h after the last administration of IMI or saline (sal). Results are expressed as a percentage of sham/sal group ± standard error of mean (SEM), n = 6; (**C**) representative blots; * *p* < 0.05 relative to sham/sal; + *p* < 0.05 relative to stress/sal group; stress/IMI resp—group of CMS rats that responded to IMI treatment (reversed anhedonia in sucrose test); stress/IMI nonresp—group of CMS rats that did not respond to IMI.

**Figure 5 ijms-25-00243-f005:**
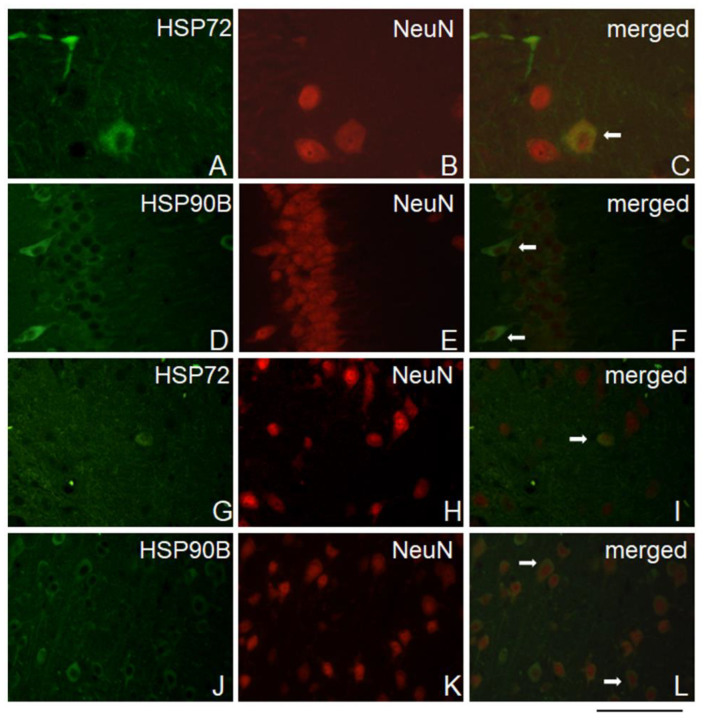
Representative double immunofluorescence staining for HSP72 or HSP90B and NeuN. Arrows indicate HSP72-positive or HSP90B-positive cells colocalized with neurons in the (**A**–**F**) PFC and (**G**–**L**) HIP areas. Scale bar 25 μm.

**Figure 6 ijms-25-00243-f006:**
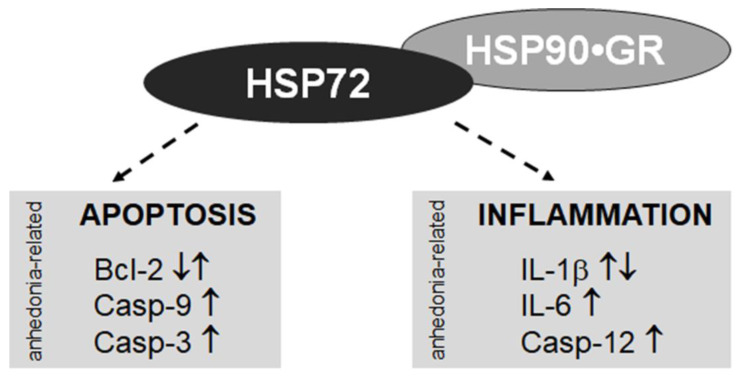
The HSP72 as a potential target of antidepressant treatment. Signaling molecules involved in the mechanism of anhedonia are shown in the rectangles. Directions of changes within the cerebral prefrontal cortex (PFC) and/or the hippocampus (HIP) of animals with anhedonia phenotype are illustrated by arrows. HSP72 working directly and/or by cooperation with HSP90 in the GR chaperone complex is the upstream regulator of apoptosis and inflammatory processes, as described in the Section 3.2.3. Therefore, modulation of HSP72 activity may normalize the apoptotic and inflammation pathways affected by depression (indicated by the dotted arrows). The HSP72 mRNA expression is increased in the PFC and HIP of imipramine-resistant CMS rats as revealed by the current study. Bcl-2, B-cell lymphoma 2 apoptosis regulator; Casp-9, Caspase 9; Casp-3, Caspase 3; Casp-12, Caspase 12; Il-1β, Interleukin 1β; Il-6, Interleukin 6; GR, glucocorticoid receptor; HSP72, Heat Shock Protein 72; HSP90, Heat Shock Protein 90 [15,47,73,74,79,80,81].

**Table 1 ijms-25-00243-t001:** Lack of changes in expression of mRNAs of HSP70 and HSP90 families in the (A) PFC, (B) HIP and (C) Thal of rats subjected to the CMS procedure (stress) for 3 weeks determined by real-time PCR method.

Brain Structure/ Treatment	Level of mRNAs [% of Sham Control ± SEM] ^1^			
	HSP72	HSC70	HSP90A	HSP90B
**A. PFC**				
Sham	100.00 ± 10.98	100.00 ± 7.91	100.00 ± 5.86	100.00 ± 7.70
Stress-reactive	93.63 ± 10.57	96.64 ± 6.89	96.84 ± 5.97	94.30 ± 5.92
Stress-nonreactive	92.76 ± 8.94	96.33 ± 3.96	100.43 ± 2.96	85.88 ± 3.80
One-way ANOVA	*F(2,15) = 0.15*, *p* > 0.05	*F(2,15) = 0.10*, *p* > 0.05	*F(2,15) = 0.15*, *p* > 0.05	*F(2,15) = 1.39*, *p* > 0.05
**B. HIP**				
Sham	100.00 ± 10.58	100.00 ± 6.83	100.00 ± 3.00	100.00 ± 8.26
Stress-reactive	89.10 ± 8.30	99.04 ± 3.88	88.06 ± 2.82 *	100.47 ± 5.64
Stress-nonreactive	81.66 ± 16.30	78.67 ± 12.15	91.93 ± 2.04	85.89 ± 13.92
One-way ANOVA	*F(2,15) = 0.57*, *p* > 0.05	*F(2,15) = 2.08*, *p* > 0.05	*F(2,15) = 5.28*, *p* < 0.05	*F(2,15) = 0.51*, *p* > 0.05
**C. Thal**				
Sham	100.00 ± 14.45	100.00 ± 7.50	100.00 ± 9.72	100.00 ± 14.09
Stress-reactive	119.54 ± 18.15	116.78 ± 3.39	127.81 ± 3.76	136.17 ± 7.84
Stress-nonreactive	112.58 ± 15.72	108.21 ± 11.09	99.51 ± 20.64	121.24 ± 19.32
One-way ANOVA	*F(2,15) = 0.31*, *p* > 0.05	*F(2,15) = 0.98*, *p* > 0.05	*F(2,15) = 1.37*, *p* > 0.05	*F(2,15) = 1.37*, *p* > 0.05

^1^ Results are expressed as a percentage of sham group ± standard error of mean (SEM); n = 6, * *p* < 0.05 relative to control (sham) group; reactive—group of CMS rats responding to stress with anhedonia in sucrose test; nonreactive—group of CMS rats that did not develop anhedonia in sucrose test.

## Data Availability

The data presented in this study are available within the article or Appendix A.

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
