# Peer review of "Elevated Expression of HSP72 in the Prefrontal Cortex and Hippocampus of Rats Subjected to Chronic Mild Stress and Treated with Imipramine"

_ijms, 2023, doi:10.3390/ijms25010243_

Round 1

Reviewer 1 Report

Comments and Suggestions for Authors

General comments

The manuscript was written very careful, and with a good knowledge of the field. In addition, the corresponding author is a very knowledge person of the field. This article is already published at Pre-prings.org  https://doi.org/10.20944/preprints202311.0276.v1, not sure if IJMS will be happy with this. But the topic of the publication is very novel and original. However, most of the data is not significant. Although the author is taking a great advantage of the few pieces of data that was significant, to get an appropriate conclusion. I suggest the author to look more into downstream proteins with some interaction with HSP70, maybe the CMS treatment might be having a higher impact on those proteins. That should increase the chances of getting a bigger picture of CMS and significant data as well.

 Particular comments

Figure 1. The animal model reacted well to the stress (CMS), also the stress non-reactive looks good. So the animal preparation was a very good designed.

In table 1, only HPS90A mRNA levels, in HIP was significant for one of the conditions.

Table 2 and 3, nothing is significant. I recommend to remove the tables and just mention them briefly in results.

Figure 2. I many times, mRNA levels might changes but not protein levels. In case of figure 2, only mRNA levels changed for “stress IMI no resp”. In my opinion "mRNA levels is not adding nothing new to the paper, but the protein levels is good, imipramine is restoring normal levels of HSP72. I recommend the author to eliminate the figure of "mRNA expression" only, and leave the western blot.

Figure 3. The "mRNA expression" figure is not adding nothing to the article. I recommend to eliminate the figure of "RNA expression only", and leave the western blot.

Figure 4. Although there is significant data for the "HSP90B mRNA levels", there is nothing for the protein levels (western blot). Again the figure is not adding nothing to the paper, I recommend to remove the figure and just mention it briefly in results.

Table 4, Nothing is significant in this table. I recommend to remove this table and just mention it briefly in results.

Figure 5. Just looking up for co-localization between HSP72 or HSP90B with NeuN is not enough. They might overlay, but might not co-localize. I will strong recommend the author to acquire Z-stacks images, which is a compilation of photographs taken at a set interval between the first and last lanes of focus, that will allow you to do a 3D image of your tissue, to really determine if the HSP72 or HSP90B are co-localized in neurons (NeuN).

Reviewer 2 Report

Comments and Suggestions for Authors

The paper evaluated the HSPs protein in two protocols of depressive model.

I belive that the most interesting results are the non-stress reactive and the IMI nonrespond groups which are almost not discuss in the manuscript. The discussion is to long. It should be more direct, discussing the results that the authors found.

The authors have others results to suport their discussion? Why only the HSPs were studied?

Round 2

Reviewer 1 Report

Comments and Suggestions for Authors

I don't have any comments, the authors addressed my concerns.

Thank you